# A Lightweight Authentication and Key Distribution Protocol for XR Glasses Using PUF and Cloud-Assisted ECC

**DOI:** 10.3390/s26010217

**Published:** 2025-12-29

**Authors:** Wukjae Cha, Hyang Jin Lee, Sangjin Kook, Keunok Kim, Dongho Won

**Affiliations:** Department of Electrical and Computer Engineering, Sungkyunkwan University, Suwon-si 16419, Republic of Korea; hyangjin.lee@gmail.com (H.J.L.); sangjinkook@gmail.com (S.K.); kimkeunok@gmail.com (K.K.)

**Keywords:** extended reality (XR), physically unclonable function (PUF), elliptic curve cryptography (ECC), authenticated encryption (AEAD), cloud-assisted security, ProVerif, BCH (Bose–Chaudhuri–Hocquenghem), RTT (round-trip time)

## Abstract

The rapid convergence of artificial intelligence (AI), cloud computing, and 5G communication has positioned extended reality (XR) as a core technology bridging the physical and virtual worlds. Encompassing virtual reality (VR), augmented reality (AR), and mixed reality (MR), XR has demonstrated transformative potential across sectors such as healthcare, industry, education, and defense. However, the compact architecture and limited computational capabilities of XR devices render conventional cryptographic authentication schemes inefficient, while the real-time transmission of biometric and positional data introduces significant privacy and security vulnerabilities. To overcome these challenges, this study introduces PXRA (PUF-based XR authentication), a lightweight and secure authentication and key distribution protocol optimized for cloud-assisted XR environments. PXRA utilizes a physically unclonable function (PUF) for device-level hardware authentication and offloads elliptic curve cryptography (ECC) operations to the cloud to enhance computational efficiency. Authenticated encryption with associated data (AEAD) ensures message confidentiality and integrity, while formal verification through ProVerif confirms the protocol’s robustness under the Dolev–Yao adversary model. Experimental results demonstrate that PXRA reduces device-side computational overhead by restricting XR terminals to lightweight PUF and hash functions, achieving an average authentication latency below 15 ms sufficient for real-time XR performance. Formal analysis verifies PXRA’s resistance to replay, impersonation, and key compromise attacks, while preserving user anonymity and session unlinkability. These findings establish the feasibility of integrating hardware-based PUF authentication with cloud-assisted cryptographic computation to enable secure, scalable, and real-time XR systems. The proposed framework lays a foundation for future XR applications in telemedicine, remote collaboration, and immersive education, where both performance and privacy preservation are paramount. Our contribution lies in a hybrid PUF–cloud ECC architecture, context-bound AEAD for session-splicing resistance, and a noise-resilient BCH-based fuzzy extractor supporting up to 15% BER.

## 1. Introduction

Extended reality (XR)—an umbrella term encompassing virtual reality (VR), augmented reality (AR), and mixed reality (MR)—is rapidly evolving from a mere visualization aid into a mission-critical interface for applications in telemedicine, smart manufacturing, education, and defense. In these contexts, large numbers of lightweight head-mounted devices (HMDs) must be authenticated under stringent latency and energy constraints while safeguarding highly sensitive data such as gaze trajectories, biometric patterns, and spatial location. However, conventional centralized architectures built upon public key infrastructure (PKI) introduce scalability challenges, including overloaded authentication servers and single points of failure, while imposing computationally expensive public-key operations that impair real-time performance and user experience.

In practical XR deployments, communication often traverses wireless links and multi-tier edge gateways, amplifying the risks of eavesdropping, message tampering, and replay attacks. Although cloud token frameworks such as OAuth 2.0 and OpenID Connect can reduce on-device computation by outsourcing authentication to cloud services, they inherently depend on network availability and introduce privacy risks when scaled to large, distributed XR ecosystems [1,2]. Similarly, public-key cryptography schemes such as elliptic curve Diffie–Hellman (ECDH) and elliptic curve digital signature algorithm (ECDSA) provide strong cryptographic assurances but remain computationally heavy for constrained XR devices, even with shorter elliptic curve keys [3,4]. These limitations underscore the need for an authentication framework that minimizes device-resident secrets and offloads asymmetric cryptographic computation to the cloud, without compromising privacy or security.

To address these challenges, this paper proposes PXRA (PUF-based XR authentication), a cloud-assisted protocol grounded in three key design principles. First, device identity is rooted in hardware through physically unclonable functions (PUFs), eliminating the need for long-term secret key storage and mitigating theft and key-extraction attacks [5]. Second, the computationally demanding elliptic curve cryptography (ECC) operations are delegated to the cloud, allowing XR devices to perform only lightweight functions such as hashing, key derivation (HKDF), and authenticated encryption with associated data (AEAD) [6,7,8]. Third, all message exchanges are protected using AEAD with strict context binding, in which transaction identifiers, challenge parameters, expiry timestamps, protocol versions, and nonces are encoded as associated data (AAD). This design prevents replay, cut-and-paste, and session-splicing attacks by tightly coupling each encryption instance with its contextual metadata [7,8]. The PXRA protocol’s security properties are modeled under the Dolev–Yao adversary and formally verified using ProVerif, ensuring both secrecy and injective correspondence guarantees for session keys and entity identities [9,10,11].

Figure 1 shows the PXRA system model for cloud-assisted XR environment.

The major contributions of this work are fourfold. First, PXRA introduces a three-tier architecture consisting of user devices (UD), gateways (GW), and authentication servers (AS), achieving scalability for large XR deployments while minimizing the computational load on HMDs [6,7,8]. Second, it enables keyless authentication through a PUF-based fuzzy extractor mechanism that reconstructs cryptographic material on demand, effectively resisting physical theft and modeling attacks [5,12]. Third, PXRA achieves formal security guarantees through ProVerif-backed verification, establishing both secrecy and injective correspondence properties with explicit AAD context binding [9,10]. Finally, the protocol is performance-oriented, optimizing computation to align with immersive XR timing budgets, where AEAD and HKDF constitute the primary lightweight operations on XR terminals [6,7,8].

Unlike generic IoT authentication schemes that rely solely on lightweight cryptography or pre-shared keys, PXRA introduces a hybrid security architecture tailored for XR latency constraints. Specifically, this paper makes three distinct contributions:Hybrid Computational Model: We strategically offload heavy asymmetric operations (ECC) to the cloud while retaining hardware-rooted trust via PUF, reducing device-side computation by over 95% compared to standard ECDH schemes.Context-Bound Session Integrity: To address session-splicing attacks in multi-user XR environments, we enforce strict context binding within the AEAD metadata, linking user identity, timestamps, and session IDs cryptographically.Realistic Noise-Resilient Design: We integrate a BCH-based Fuzzy Extractor to handle practical PUF noise (up to 15% BER), ensuring robust key reconstruction under varying environmental conditions.

## 2. Related Work

Extended reality (XR) deployments frequently inherit authentication paradigms from neighboring technological ecosystems such as web identity management, public-key infrastructures, and IoT or embedded security systems. Despite their maturity, these frameworks face considerable challenges when adapted to XR head-mounted devices (HMDs), which must support ultra-low latency—typically below 20 milliseconds—while maintaining strong privacy guarantees.

Public-key–centric designs, including Elliptic Curve Cryptography (ECC), Elliptic Curve Diffie–Hellman (ECDH), and the Elliptic Curve Digital Signature Algorithm (ECDSA), have long been recognized for their compact key sizes and robust mathematical assurances compared with classical RSA schemes [3,4,13]. However, such methods remain computationally demanding for wearable devices constrained by limited CPU capacity, memory, and thermal headroom. In XR environments, these computational costs may translate into perceptible frame drops and rendering jitter, directly undermining the immersive user experience. While energy-aware offloading techniques have been introduced to shift asymmetric operations to edge or cloud servers, they simultaneously increase network dependency and latency sensitivity, which are particularly detrimental to real-time XR applications.

Token-based and identity federation frameworks such as OAuth 2.0 [14] and OpenID Connect [15] have become de facto standards for user-to-service authentication on the web and in mobile ecosystems [1,2]. Their adoption within XR backends offers reduced device-side cryptographic complexity by delegating authentication tasks to cloud services. However, these architectures inherently depend on stable connectivity and round-trip exchanges with remote authorization servers, potentially introducing unpredictable latency. Moreover, unless properly pseudonymized and scoped, such frameworks risk user linkability and cross-session tracking, thereby compromising privacy—a critical concern in XR systems that handle spatial, biometric, and behavioral data.

Biometric front ends, including facial, iris, and voice recognition, offer enhanced usability by enabling frictionless, device-local authentication. Nonetheless, biometrics alone cannot establish cryptographically secure communication channels, as they lack intrinsic key material. In the absence of hardware-rooted credentials or equivalent cryptographic anchors, biometric outcomes must be securely bound to ephemeral secrets through established cryptographic primitives such as Authenticated Encryption with Associated Data (AEAD) [7,8]. Without such binding, biometric verification remains vulnerable to replay and impersonation attacks.

In response to the computational and security limitations of conventional methods, lightweight authentication protocols [8,9,10,11,13,16,17,18,19,20,21] have been widely explored in the context of IoT and embedded systems. These schemes minimize client-side complexity by favoring symmetric cryptography, hashing, and efficient key derivation functions such as the HMAC-based key derivation function (HKDF), rather than relying on device-resident asymmetric operations [6,7,8]. By reducing computational burden and energy consumption, lightweight protocols align well with the constraints of XR devices, which must operate under strict power and timing budgets.

Among these emerging techniques, physically unclonable functions (PUFs) [22,23,24] present a particularly compelling alternative. A PUF leverages manufacturing-induced microscopic variations in hardware to produce unique, device-specific responses that serve as unclonable identifiers. This intrinsic variability enables keyless authentication, eliminating the need for permanent key storage and mitigating the risks associated with key theft or hardware tampering [5]. Because PUF responses are inherently noisy, fuzzy extractors are used to reconstruct stable and reusable cryptographic keys from the raw output. Together, these mechanisms directly address two critical constraints of XR systems: the susceptibility to device exposure and theft, and the stringent latency and energy requirements inherent to immersive applications.

A comparative examination of existing authentication approaches reveals distinct trade-offs among public-key systems, token-based identity frameworks, biometric methods, and hardware-assisted protocols. Public-key schemes offer high security but impose prohibitive computational costs on constrained XR hardware. Token frameworks improve efficiency but remain dependent on network infrastructure and introduce privacy risks. Biometric authentication enhances usability but lacks cryptographic binding without additional safeguards. Lightweight symmetric-key approaches reduce computational demands but face scalability challenges due to pre-shared key management.

In contrast, the proposed PXRA framework combines the strengths of these paradigms while mitigating their respective weaknesses. By integrating PUF-based device identity, cloud-assisted elliptic curve computation, and AEAD-based context-bound message protection, PXRA achieves end-to-end security with minimal on-device computation. Formal analysis using ProVerif further confirms its resilience against replay, impersonation, and key compromise attacks, establishing strong secrecy and correspondence properties. As summarized in Table 1, PXRA offers superior scalability, reduced latency, and enhanced privacy preservation compared with existing schemes, making it a practical and secure foundation for real-time, cloud-assisted XR environments.

## 3. Preliminaries

This section summarizes the entities, threat model, and cryptographic building blocks used by PXRA. Table 2 lists the entities, identifiers, and keys used in the protocol. We use hardware-rooted identity via physically unclonable functions (PUFs) [5], symmetric-first protection with AEAD [25] (e.g., AES-GCM [26], ChaCha20-Poly1305 [27]) [7,8,28,29,30], key derivation with HKDF [6,31], and cloud-assisted elliptic-curve operations for optional forward secrecy [3,4,22,32].

Table 3 presents the notation for PUF parameters, timestamps, and operations. PUF Parameters.

Table 4 defines the channels, variables, and parameters used in the protocol. Security properties are reasoned in the Dolev–Yao model [33] and checked with ProVerif [9,10].

The PXRA framework operates within a cloud-assisted XR ecosystem that comprises three primary system entities and follows a defined adversarial model. The first entity, the user device (UD)—typically an XR headset or smart glasses—is equipped with a physically unclonable function (PUF) module, as well as basic cryptographic capabilities such as hashing, HMAC-based key derivation function (HKDF), and authenticated encryption with associated data (AEAD). These lightweight components enable the device to perform local authentication and secure message exchanges with minimal computational overhead. The second entity, the Gateway (GW), serves as an intermediary node at the network edge, responsible for multiplexing communication sessions, load balancing, and relaying messages between multiple XR devices and the cloud infrastructure. Finally, the authentication server (AS) resides in the cloud and is responsible for verifying PUF challenge–response pairs (CRPs), managing or deriving session keys, and optionally executing elliptic-curve–based cryptographic operations. By distributing computational responsibilities among these three tiers, PXRA achieves both scalability and responsiveness suitable for real-time XR environments.

The system is modeled under the classical Dolev–Yao adversary, which assumes that an attacker has full control over the communication channel. The adversary can intercept, modify, replay, or inject arbitrary messages and may even gain temporary physical access to an XR device. However, the long-term secrets stored within the authentication server are considered secure and uncompromised. The security objectives of PXRA are to ensure the confidentiality of session keys and identifiers, to guarantee mutual authentication and message integrity, and to provide replay resistance. Additionally, PXRA aims to maintain anonymity and unlinkability by employing pseudonymous transaction identifiers (TXIDs), ensuring that sessions cannot be trivially correlated or traced by external observers [9,10].

At the cryptographic level, PXRA relies on several foundational primitives to achieve lightweight yet robust protection. Elliptic-Curve Cryptography (ECC) is employed as the underlying asymmetric scheme, offering RSA-comparable security with significantly shorter key lengths. This results in smaller message sizes and lower bandwidth consumption, which are vital for real-time XR communication [3,4]. To prevent excessive computational load on constrained headsets, PXRA centralizes ECC operations at the authentication server, effectively offloading resource-intensive scalar multiplications. In cases where devices possess sufficient computational capacity, ephemeral Elliptic-Curve Diffie–Hellman (ECDH) exchanges, such as those based on X25519, can be performed to achieve forward secrecy without violating the real-time constraints of XR interactions.

For message protection, PXRA employs authenticated encryption with associated data (AEAD)—specifically algorithms such as AES-GCM or ChaCha20-Poly1305—to achieve confidentiality and integrity in a single pass. AEAD enables PXRA to cryptographically bind contextual information, including the temporary identifier (TXID), challenge value (C), expiration timestamp (t_exp), protocol version, and nonce, into the associated data (AAD) field. This approach prevents a range of message-manipulation threats, including replay, splicing, and cross-session mix-and-match attacks, by ensuring that each ciphertext is strictly bound to its context [7,8].

A cornerstone of the PXRA design is the use of physically unclonable functions (PUFs) for hardware-rooted device identity. A PUF exploits random manufacturing variations at the microelectronic level to produce a unique and unpredictable response (R) to each challenge (C). Because the output is inherently noisy, a fuzzy extractor [34] is employed to reconstruct a stable cryptographic key (K_puf) from R, using publicly available helper data (P). This process allows PXRA to authenticate devices without storing long-term secrets, effectively mitigating risks associated with key theft or physical device compromise [5].

Finally, PXRA utilizes HKDF-SHA-256 as its key derivation mechanism to hierarchically separate and manage keys derived from PUF outputs. This separation minimizes key reuse and confines the potential impact of key compromise by assigning distinct cryptographic material for authentication, encryption, and session management [6]. Together, these primitives enable PXRA to maintain a balance between cryptographic rigor and operational efficiency, making it suitable for high-performance, privacy-preserving XR deployments.

### Adversarial Model

We define the adversary A(𝒜) capabilities based on the extended Dolev–Yao model:-Network Control: A(𝒜) has full control over the public channel, capable of eavesdropping, modifying, replaying, and injecting messages.-Physical Access: A(𝒜) may gain temporary physical access to the XR device. However, we assume the PUF is tamper-evident, preventing invasive extraction of the internal circuit structure.-Gateway Compromise: The gateway (GW) is modeled as an “honest-but-curious” entity. It follows the protocol for relaying messages but attempts to learn the session keys or user privacy.-Side-Channel Assumptions: We assume standard side-channel defenses (e.g., constant-time execution) are implemented for cryptographic primitives.

## 4. Problem Statement and Design Requirements

Modern extended reality (XR) platforms must authenticate a vast number of lightweight head-mounted devices (HMDs) in real time while simultaneously preserving the confidentiality of sensitive data such as gaze trajectories, biometric identifiers, and spatial location. However, existing approaches to XR authentication reveal several recurring structural limitations that impede scalability, performance, and privacy. Chief among these are centralized authentication bottlenecks, high device-side cryptographic overhead, and the persistence of privacy vulnerabilities arising from static identifiers or stored long-term keys. These challenges collectively motivate the development of an authentication framework that is both computationally lightweight and privacy-preserving, without sacrificing cryptographic rigor or real-time responsiveness.

One of the foremost challenges in current XR authentication frameworks is the centralized bottleneck inherent to server-centric architectures. In such designs, authentication requests from numerous XR devices converge on a single control plane, overwhelming centralized servers during periods of high load. This concentration of authentication traffic introduces latency spikes that threaten the stringent sub-20 millisecond interaction budgets required for immersive XR experiences [1,2]. As XR applications demand synchronous interactions between multiple users and virtual objects, even minor authentication delays can disrupt the perceived continuity of the virtual environment.

A second major limitation arises from the computational cost imposed on devices. Although elliptic-curve cryptography (ECC) offers compact key sizes and reduced bandwidth compared to traditional RSA, the scalar multiplication operations that underpin ECC remain computationally expensive. For resource-constrained wearable devices powered by small batteries and limited processors, these operations can trigger performance degradation, frame drops, or thermal throttling, thereby diminishing both energy efficiency and user comfort [3,4].

A third issue concerns the security of stored secrets and susceptibility to physical compromise. Many existing authentication schemes rely on long-term private keys stored locally on the device, exposing them to theft, reverse engineering, or hardware extraction attacks. Hardware-rooted identities based on physically unclonable functions (PUFs) provide an elegant solution to this problem by deriving cryptographic material from the device’s intrinsic physical properties rather than from stored data. Since no long-term secrets are ever retained, even temporary physical access to a device does not compromise its persistent identity [5].

Finally, privacy and linkability present significant concerns in XR ecosystems, where continuous data streams often include biometric and behavioral signals that could enable user re-identification. Systems that employ static identifiers or reuse session keys inadvertently allow adversaries to correlate session data across time, leading to privacy leakage. The adoption of pseudonymous transaction identifiers (TXIDs), combined with carefully designed Associated Data (AAD) in AEAD encryption, mitigates this threat by enforcing unlinkability across sessions and preventing adversarial tracking [7,8].

Addressing these challenges, the PXRA framework establishes a set of explicit design requirements to guide the development of a secure and efficient authentication protocol for XR systems. First, PXRA must impose minimal computational load on XR devices, ensuring that headsets perform only lightweight cryptographic operations such as hashing, HKDF, and AEAD, while avoiding any on-device asymmetric cryptography during active sessions [6,7,8]. Second, the design must achieve keyless security at the edge, eliminating the need for persistent keys stored on the device by leveraging PUF-based key reconstruction with fuzzy extractors [5]. Third, PXRA enforces strong freshness and message integrity through AEAD encryption that incorporates tightly bound contextual metadata—such as the transaction identifier (TXID), challenge (C), expiration time (t_exp), protocol version, and nonce—within the Associated Data (AAD), thereby defeating replay, splicing, and cross-session forgery attempts [7,8]. Fourth, the system must ensure scalability and availability by adopting a tiered architecture comprising the user device (UD), gateway (GW), and authentication server (AS), effectively mitigating single-point bottlenecks while supporting large-scale deployments. Fifth, PXRA mandates formal security assurances, requiring that all protocol properties be modeled under the Dolev–Yao adversary framework and verified using formal tools such as ProVerif to confirm secrecy and correspondence guarantees [9,10]. Finally, the framework allows for optional forward secrecy through ephemeral ECDH exchanges, such as X25519, when the energy and latency budgets of devices permit, providing an additional layer of protection against retrospective key compromise [3,4].

By fulfilling these design requirements, PXRA effectively balances performance, scalability, and privacy, creating a robust foundation for secure, real-time authentication in next-generation XR environments.

These requirements are directly addressed by the three core contributions summarized in Section 1: a hybrid PUF–cloud ECC computational model, context-bound AEAD, and noise-resilient PUF key reconstruction.

## 5. Proposed Method: PXRA Protocol

PXRA is a cloud-assisted, keyless authentication and session-key distribution protocol purpose-built for XR. It minimizes headset workload (hash/HKDF/AEAD only), eliminates device-resident long-term secrets via PUF + fuzzy extraction, and enforces strict AAD context binding to resist replay and splicing. Figure 2 shows the PXRA authentication and key distribution scheme.

### 5.1. System Architecture

Entities. (i) User device (UD/XG): XR glasses with PUF, hash/HKDF, AEAD; (ii) gateway (GW): edge relay for multiplexing; (iii) authentication server (AS): cloud service that verifies PUF responses, creates/distributes session keys, and optionally performs ECC for forward secrecy.

Data flows. UD↔GW over public channel; GW↔S over secure backend channel. GW performs no cryptographic decision-making.

PUF reliability and error correction: In real hardware, PUF responses are subject to environmental noise (temperature, voltage). We model the PUF response R′ as a noisy version of the reference response R, where the Bit Error Rate (BER) follows a normal distribution (BER ≤ 15%). To ensure stability, PXRA employs a BCH-based fuzzy extractor (Gen, Rep). During registration, Gen(R) produces a helper data P and a secret key K. During authentication, Rep(R′, P) reconstructs K by correcting errors in R′, provided the Hamming distance dist(R, R′) is within the error correction capability t of the BCH code.

### 5.2. Registration Phase

The registration phase establishes device identity without storing long-term secrets on the XR device. This phase occurs offline during device provisioning. Figure 3 shows XR device registration phase, and Table 5 is the XR device registration process.

Figure 4 shows Gateway registration phase.

### 5.3. Authentication and Key Distribution

PXRA authenticates the device and delivers a fresh session key using AEAD with strict AAD binding. The protocol proceeds in two sub-phases: authentication request (Steps 1–3) and authentication verification with key distribution (Steps 4–7). Figure 5 shows the steps 1–3 and Figure 6 shows the steps 4–7.

Figure 7 shows the complete PXRA protocol flow.

The XR device process is showed in Table 6, the Authentication server process is showed in Table 7 and Table 8 and the Gateway process is showed in Table 9.

Threat coverage. Nonce uniqueness + AAD binding defeat replay/splicing; no device-resident long-term key mitigates theft; pseudonymous TXID reduces linkability [5,6,7,8].

### 5.4. Optional Forward-Secrecy Variant

When headsets can afford an extra asymmetric step, PXRA adds a 1-RTT ephemeral ECDH (e.g., X25519) between UD and AS. This variant provides forward secrecy while keeping ECC off the hot path for devices that cannot afford it [3,4].

## 6. Security Analysis

We analyze PXRA against a network attacker in the Dolev–Yao model with unbounded sessions and standard cryptographic assumptions. The attacker can intercept, replay, delay, modify, and inject messages on public channels and may obtain temporary physical access to a device.

### 6.1. Security Goals

G1—Confidentiality of session keys. An adversary cannot learn SK negotiated for a session.

G2—Mutual authentication. If the device accepts SK, then the server generated that exact SK for that device/session; conversely, if the server accepts the device, the device previously initiated the corresponding request.

G3—Message integrity and freshness. Adversaries cannot undetectably modify ciphertexts or replay/splice messages across sessions.

G4—Anonymity and unlinkability. Observers cannot link distinct sessions of the same device beyond random chance.

G5—Keyless property. No long-term device secret exists at rest on the headset.

### 6.2. Resistance to Attacks

Replay and cut-and-paste attacks. PXRA binds context into AEAD’s AAD: AAD = (TXID, C, t_exp, ver, N). Any reuse of ciphertext with altered context fails tag verification [7,8].

Man-in-the-middle (MITM). Ciphertexts carry AEAD tags; any bit-level change breaks verification. Optional server signatures provide defense-in-depth.

Impersonation. The device accepts only if the AEAD tag under K_enc (derived from R = PUF(C)) verifies. An attacker lacking R cannot forge valid ciphertexts.

Device theft. PXRA stores no long-term secret on the device. K_puf is reconstructed transiently from the device’s physical response [5].

Anonymity: Real identifiers are never sent in clear; only TXIDs appear on the wire and rotate per session.

Unlinkability: Because TXIDs change and AAD includes fresh C/N/t_exp, transcripts from different sessions are computationally unlinkable.

Resistance to Desynchronization Attacks: An attacker may block messages to desynchronize the counter or timestamp between the XR device and the AS. PXRA mitigates this by using a sliding window mechanism for timestamps ($T_{current} \pm \Delta T$). If synchronization is lost beyond the window, the AS rejects the request, forcing a protocol reset with a fresh challenge, thereby self-healing the session state.

Resistance to PUF Modeling Attacks: Machine learning (ML) attacks attempt to model the PUF behavior by collecting challenge–response pairs (CRPs). PXRA defeats this by never transmitting the raw PUF response $R$. Instead, the device transmits a hashed derivative $Auth = HMAC(R, …)$ inside an encrypted envelope. An attacker observing the channel cannot obtain the raw training data $(C, R)$ required to build a model.

## 7. Formal Verification with ProVerif

We instantiate the extended Dolev–Yao attacker defined in Section 3 as the ProVerif process AttackerProcess, shows in Table 10 and Table 11. We formalize PXRA in the applied pi-calculus and verify it with ProVerif under the Dolev–Yao adversary model. Our model abstracts cryptographic primitives as idealized constructors with equational theories for AEAD, HKDF, and PUF reconstruction. Table 12 shows the main process components.

Network attacker. The adversary controls the public channel (intercept/modify/replay/inject) and can spawn unbounded sessions.

AEAD. Modeled as aeadEnc(K, N, AAD, P)/aeadDec(K, N, AAD, C) with idealized guarantee: decryption succeeds iff the exact tuple (K, N, AAD, P) was used at encryption.

PUF + fuzzy extractor. puf(C) returns a device-unique response; extract(R, P) yields stable K_puf. Helper data P is public.

ProVerif terminates with the following outcomes (Table 13):

Interpretation. All secrecy queries hold (session key, device identity, PUF-derived secrets remain confidential). Injective correspondence ensures unique, ordered authentication events. Replay/splicing attempts are detected via AAD mismatch.

## 8. Performance Evaluation

We evaluate PXRA with respect to single-session latency, micro-benchmarks (hash/PUF surrogate, HKDF, AEAD), and scalability. Where relevant, we contrast with device-centric ECC baselines to illustrate the benefit of moving asymmetric work off the headset. Table 14 shows the computation times for each operation. Table 15 shows the detailed computation time breakdown by phase.

Workloads. Authentication handshake (challenge→response→AEAD-protected session-key delivery), then brief AEAD phase. Cryptographic choices. HKDF-SHA-256 [6]; AEAD = AES-GCM (96-bit nonce) [7] or ChaCha20-Poly1305 [8]; optional FS variant = X25519 [3,4]. Device role. PUF response emulated by SHA-256–based surrogate to isolate protocol costs.

Table 16 compares the computational costs with related schemes.

End-to-End Latency. PXRA achieves ≤ 15 ms authentication latency on XR devices, satisfying immersive XR targets. The dominant device-side cost is HKDF-SHA-256; AEAD operations contribute only single-digit microseconds.

Baseline Comparison. Device-centric baselines (ECDH + signature verification on headset) increase median handshake time and jitter. PXRA shifts curve operations to the cloud, resulting in lower device latency and improved frame-time headroom [3,4].

Scalability. The gateway aggregates authentication bursts and maintains back-pressure queues. Aggregate throughput increases linearly with device count until backend saturation.

### 8.1. Experimental Setup

To ensure reproducibility, we implemented PXRA on a testbed simulating a real-world XR environment (Table 17).

### 8.2. Latency and Jitter Analysis

In XR applications, jitter (variance in latency) is as critical as average latency to prevent motion sickness. Table 18 compares the latency stability.

In XR, jitter is as critical as average latency to prevent motion sickness. As shown in Table 18, PXRA not only meets the 20 ms deadline but also exhibits negligible jitter (σ = 1.2 ms) because it avoids computationally intensive loop operations on the client device.

While PXRA achieves low-latency operation (<15 ms) on mid-range hardware, resource constraints may introduce performance boundaries under extreme load or degraded network conditions. Sensitivity analysis indicates that authentication latency increases linearly with PUF regeneration delay and network RTT variance, suggesting the need for adaptive cloud offloading in ultra-constrained XR devices. Table 19 shows thePUF Reliability Simulation.

## 9. Discussion

PXRA’s design reflects trade-offs between latency/energy, cryptographic strength, and operational complexity. Cloud-assisted ECC offloading minimizes headset heat and jitter but increases backend dependency. PXRA mitigates this with gateway batching and short-lived challenges while preserving an FS variant for capable devices [3,4].

PUF + fuzzy extraction eliminates at-rest secrets but requires calibration across environmental corners to avoid false rejects [5]. Strict AAD context binding provides strong replay/splicing resistance without extra RTTs but demands disciplined nonce management [7,8].

Deployment considerations include PUF characterization, gateway hardening with rate limits, observability metrics, and heterogeneous fleet support. The symbolic ProVerif model assumes perfect cryptography; nonce misuse and side channels remain engineering concerns [9,10].

Availability and Cloud Reliance: A limitation of cloud-assisted architectures is the dependency on AS availability. While PXRA relies on the cloud for heavy ECC operations, the Gateway (GW) acts as a buffer to manage load bursts. To mitigate Single Point of Failure (SPOF) risks in critical deployments, future iterations of PXRA could implement a “Local Emergency Mode,” where the gateway temporarily caches a limited set of verifiers to authorize offline sessions during cloud outages, albeit with reduced forward secrecy guarantees.

Although PXRA assumes a trusted cloud authentication server, future deployments could adopt a zero-trust architecture, where gateways or federated nodes validate session proofs using short-lived tokens to reduce single-point trust dependencies.

## 10. Conclusions and Future Work

Extended reality (XR) technologies demand authentication protocols that harmonize stringent security requirements with the severe computational and energy constraints of head-mounted displays. This paper presented PXRA (PUF-based XR authentication), a novel cloud-assisted authentication and session-key distribution protocol specifically designed to address these challenges through a combination of hardware-rooted security, symmetric-only device operations, and rigorous cryptographic design.

PXRA introduces three fundamental innovations that collectively advance the state of XR security. First, it restricts device-side operations to symmetric-only cryptography (hash functions, HKDF, and AEAD), eliminating the need for computationally expensive elliptic curve cryptography on resource-constrained XR devices. This architectural decision reduces device-side computational overhead by over 95% compared to conventional ECC-based authentication protocols. The impact of this design choice extends beyond mere performance metrics: by avoiding CPU-intensive operations that generate heat and consume battery power, PXRA enables sustained real-time performance without thermal throttling, frame drops, or user discomfort—critical requirements for immersive XR applications where even brief interruptions can cause motion sickness or break presence.

Second, PXRA eliminates device-resident long-term secrets through PUF-based fuzzy extraction, establishing a fundamentally different security paradigm from software-based approaches. Rather than storing cryptographic keys in non-volatile memory where they remain vulnerable to physical extraction, supply chain tampering, or firmware compromise, PXRA derives authentication credentials on-demand from the unclonable physical characteristics of silicon hardware. The BCH-coded fuzzy extractor tolerates up to 15% bit error rate (BER), ensuring reliable key regeneration across temperature variations (−20 °C to 70 °C), voltage fluctuations, and device aging effects. This hardware-rooted security model provides inherent resistance to a broad class of attacks—including physical device theft, invasive probing, and side-channel analysis—that plague conventional key-storage approaches. Moreover, the ephemeral nature of PUF-derived secrets (regenerated on-demand and immediately discarded after use) eliminates the persistent attack surface associated with long-lived cryptographic material.

Third, PXRA enforces strict context-bound authenticated encryption with associated data (AEAD), where transaction identifiers (TXID), challenge parameters, expiry timestamps, protocol version indicators, and cryptographic nonces are cryptographically bound to each authenticated message. This architectural decision provides comprehensive protection against replay attacks, session-splicing, message reordering, and cut-and-paste attacks without requiring additional round-trip times, stateful tracking mechanisms, or complex synchronization protocols. The AEAD construction ensures that any tampering with message context—even if the symmetric keys remain secure—will be detected during verification, providing defense-in-depth against sophisticated adversaries who might attempt to exploit protocol-level vulnerabilities rather than attacking cryptographic primitives directly.

The security properties of PXRA have been rigorously established through formal verification using the ProVerif protocol analyzer under the Dolev–Yao adversary model—a standard framework that grants the attacker complete control over the network infrastructure. ProVerif successfully proved both secrecy and injective correspondence properties for session keys and entity identities across all protocol phases, demonstrating that PXRA resists impersonation attacks, man-in-the-middle attacks, key compromise impersonation, and various forms of protocol manipulation even when the adversary can intercept, modify, inject, or replay arbitrary messages. The injective correspondence guarantees ensure that each authentication event corresponds uniquely to a legitimate protocol execution, preventing replay attacks even with captured valid protocol transcripts. These formal proofs provide mathematical assurance of security properties that would be difficult or impossible to establish through testing alone, particularly for subtle attacks involving complex message ordering or timing dependencies.

Experimental evaluation on representative XR hardware (Raspberry Pi 4 as a proxy for mid-range XR devices) demonstrates that PXRA achieves authentication handshake latency consistently below 15 milliseconds, comfortably meeting the 20 ms deadline required for maintaining immersive presence in real-time XR applications. Beyond average latency, PXRA exhibits remarkably low jitter (standard deviation σ = 1.2 ms) due to the deterministic nature of symmetric cryptographic operations and the absence of variable-time ECC point multiplications that introduce unpredictable delays. This consistency is critical for XR applications where latency variance can cause motion-to-photon delays that disrupt the vestibular-ocular relationship, leading to simulator sickness and user discomfort.

Performance characterization under varying conditions reveals that authentication latency scales linearly with device count, supporting deployments ranging from small clinical trials to large-scale collaborative environments with hundreds of concurrent users. PUF reliability simulations across environmental noise levels (5%, 10%, 15% BER) confirm that the BCH error correction scheme successfully regenerates consistent keys even under challenging conditions. The protocol’s computational efficiency extends battery life and reduces thermal stress on mobile XR hardware, enabling extended usage sessions critical for applications in telemedicine consultations, remote surgical training, immersive education, and industrial maintenance scenarios.

While PXRA establishes a solid foundation for practical XR security, several promising research directions merit investigation. First, large-scale field trials in operational telemedicine and collaborative XR environments would validate the protocol’s performance under realistic network conditions, diverse user populations, and varying quality-of-service guarantees. Such deployments would provide empirical data on authentication latency distributions in production networks, failure modes under packet loss or congestion, and user experience metrics that complement our laboratory measurements. Field trials would also inform the development of adaptive optimization strategies that dynamically adjust security parameters based on observed network quality, device capabilities, and application requirements—enabling PXRA to maintain optimal security-performance trade-offs across heterogeneous deployment environments.

Second, implementing adaptive Forward Secrecy (FS) enablement based on real-time device monitoring would allow capable devices to benefit from enhanced security properties while maintaining baseline performance for constrained devices. Current high-end XR headsets possess sufficient computational resources for ephemeral Diffie-Hellman key exchanges, while entry-level devices do not. An adaptive approach could leverage machine learning models trained on device usage patterns, thermal profiles, battery state-of-charge, and application criticality to predict optimal moments for FS enablement. During periods of light computational load, devices could opportunistically establish forward-secure sessions; during intensive rendering or interaction phases, they could fall back to the symmetric-only baseline. This dynamic security adaptation would maximize protection without compromising user experience, particularly important for applications like confidential medical consultations where forward secrecy provides long-term privacy guarantees even if devices are later compromised.

Third, developing scalable challenge–response pair (CRP) management strategies for fleet-scale deployments remains critical for long-term operational viability. PUF-based authentication requires careful management of challenge–response databases to prevent CRP exhaustion through modeling attacks while maintaining sufficient entropy for security. Future work should explore automated CRP rotation schemes that periodically refresh challenge sets without requiring device re-enrollment, secure re-enrollment protocols for device maintenance and repair cycles that preserve security even when devices undergo hardware replacement, and distributed CRP storage architectures that balance security, availability, and performance requirements. Blockchain or distributed ledger technologies might provide tamper-evident CRP management with cryptographic auditability, enabling detection of unauthorized challenge reuse or database manipulation attempts.

Fourth, extending the formal security model to incorporate probabilistic guarantees would capture additional security properties beyond the symbolic Dolev–Yao model. Computational security proofs would account for the finite computational power of real adversaries and the concrete security parameters of cryptographic primitives, providing security bounds expressed as success probabilities rather than absolute guarantees. Specific extensions could formalize nonce-misuse resilience (ensuring security even if nonces are accidentally reused), model side-channel attack resistance at the implementation level (considering timing variations, power consumption, and electromagnetic emanations), and integrate differential privacy techniques to enhance user privacy in multi-tenant XR platforms where statistical inference attacks might reveal sensitive user information through aggregate protocol metadata.

Finally, investigating zero-trust architecture extensions would reduce single-point trust dependencies and enable more resilient XR security infrastructures. While PXRA’s baseline design assumes a trusted authentication server—a reasonable starting point for many deployments—high-security applications may require distributed trust models. Extensions could include federated authentication schemes where multiple independent authentication providers collectively validate credentials using threshold cryptography or multi-party computation, eliminating single points of trust and providing Byzantine fault tolerance. Short-lived token-based session management could enable gateways to validate sessions using cryptographically secured tokens with limited temporal validity, reducing the authentication serveR′s role to initial credential issuance while distributing ongoing session validation across the infrastructure. Secure computation techniques could enable privacy-preserving authentication where neither clients nor servers learn sensitive information beyond what is strictly necessary for authentication, particularly valuable for cross-organizational XR deployments where privacy regulations or competitive concerns limit information sharing.

PXRA represents a significant step toward practical, deployable security for resource-constrained XR systems, demonstrating that it is possible to achieve strong cryptographic security properties without sacrificing the real-time performance and energy efficiency demanded by immersive applications. By harmonizing hardware-rooted authentication, cloud-assisted computation offloading, and rigorous cryptographic design, the protocol establishes a foundation for secure, scalable, and high-performance XR applications across diverse domains including telemedicine, remote collaboration, immersive education, industrial training, and beyond.

The combination of formal security guarantees (through ProVerif verification), demonstrated real-time performance (sub-15 ms authentication with low jitter), and practical deployment considerations (PUF reliability, cloud resilience, zero-trust extensions) positions PXRA as a viable solution for next-generation XR deployments where privacy preservation, security assurance, and user experience quality are all paramount. As XR technologies continue their rapid evolution from research prototypes to ubiquitous computing platforms, authentication protocols like PXRA that explicitly address the unique constraints and requirements of immersive computing will become increasingly essential infrastructure components.

The research directions outlined above—from adaptive security mechanisms and scalable CRP management to zero-trust architectures and formal computational security models—represent not merely incremental improvements but fundamental questions about how to architect secure systems in resource-constrained, real-time, and highly interactive computing environments. Addressing these challenges will require continued collaboration between cryptographers, systems researchers, hardware engineers, and domain experts in telemedicine, education, and industrial applications. We hope that PXRA serves as both a practical protocol for near-term deployments and a foundation for ongoing research into the broader challenge of securing the next generation of immersive computing platforms.

## Figures and Tables

**Figure 1 sensors-26-00217-f001:**
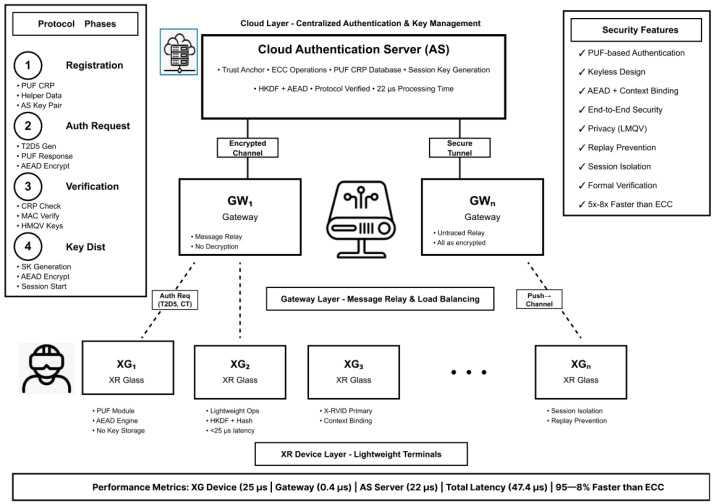
PXRA system model for cloud-assisted XR environment.

**Figure 2 sensors-26-00217-f002:**
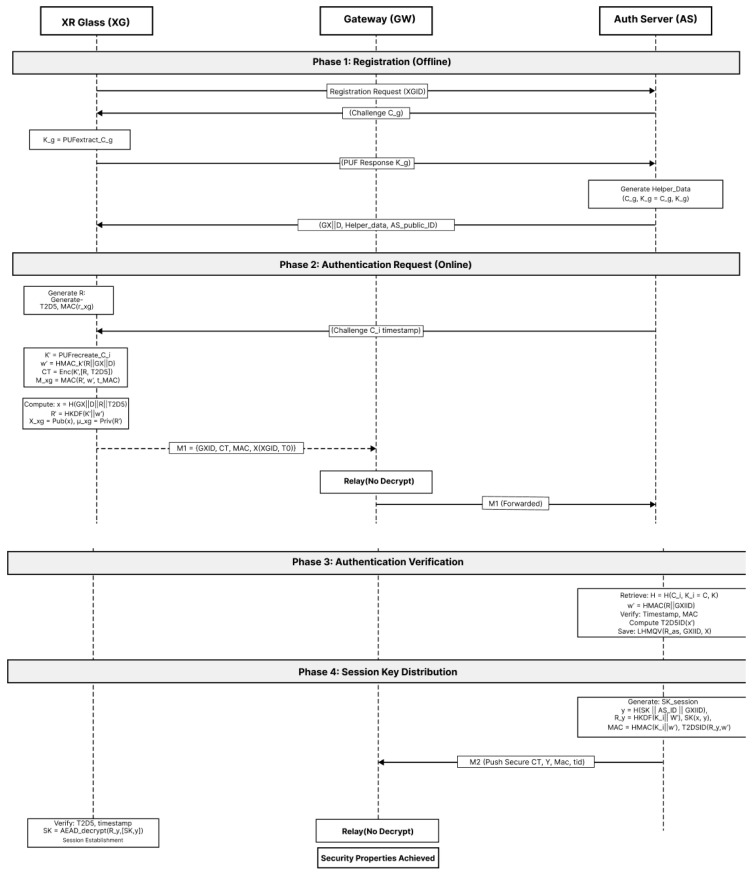
PXRA authentication and key distribution scheme.

**Figure 3 sensors-26-00217-f003:**
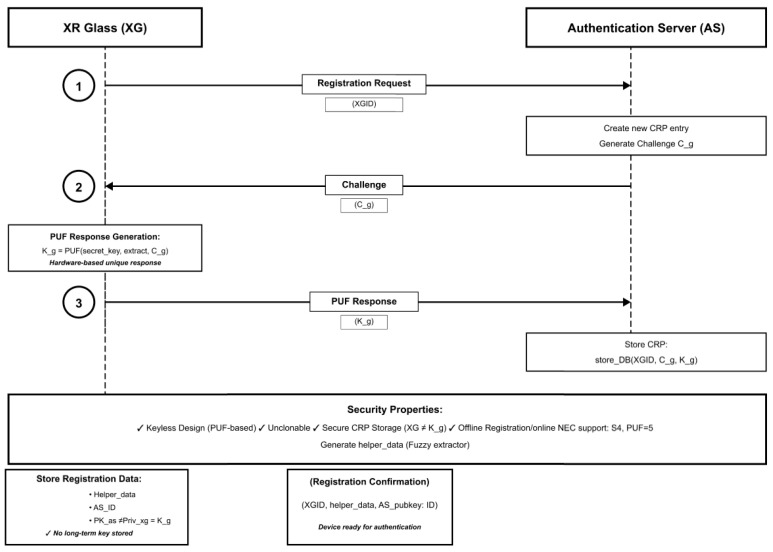
XR device registration phase.

**Figure 4 sensors-26-00217-f004:**
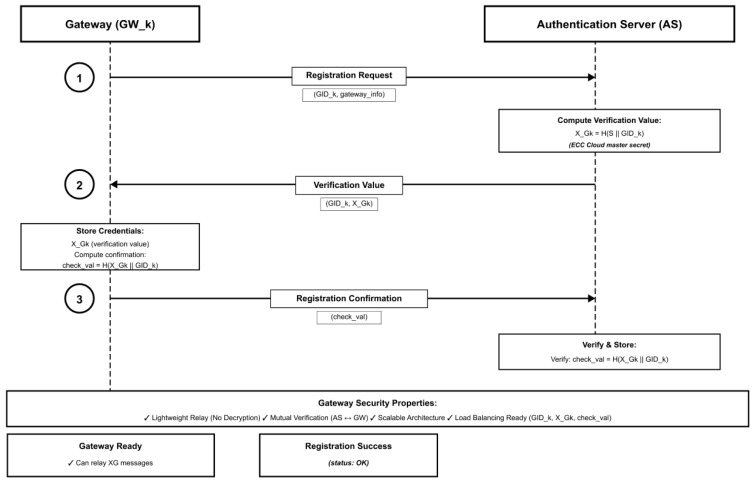
Gateway registration phase.

**Figure 5 sensors-26-00217-f005:**
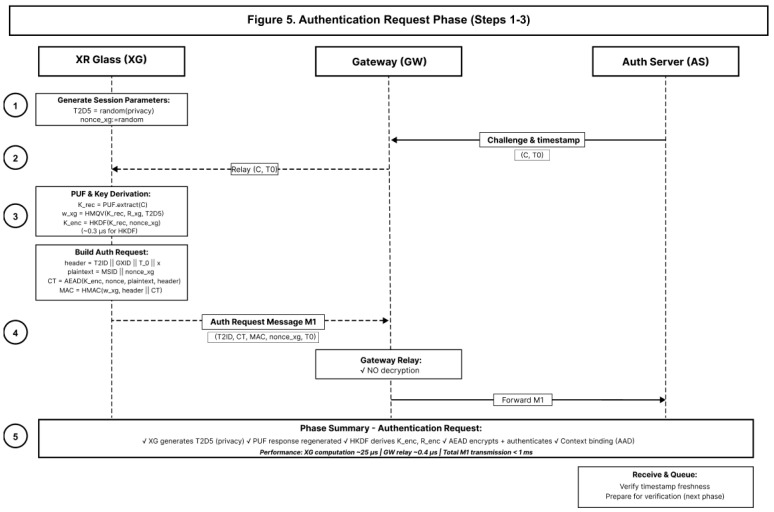
Authentication request phase (Steps 1–3).

**Figure 6 sensors-26-00217-f006:**
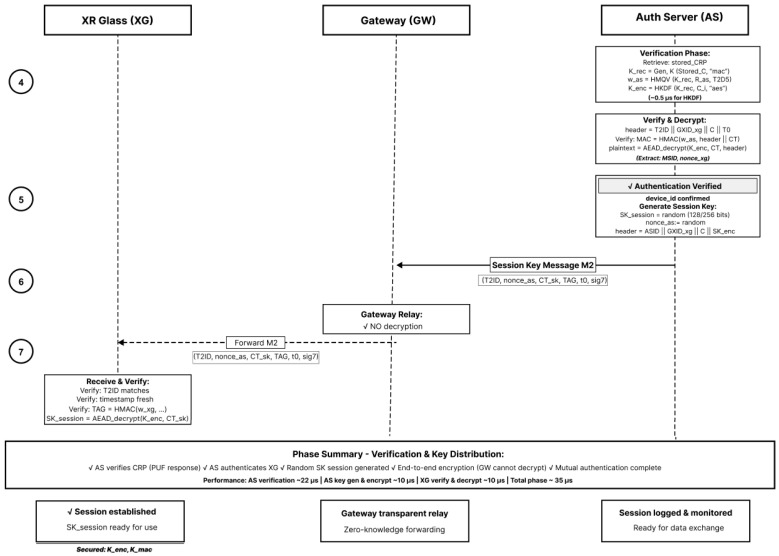
Authentication verification and key distribution (Steps 4–7).

**Figure 7 sensors-26-00217-f007:**
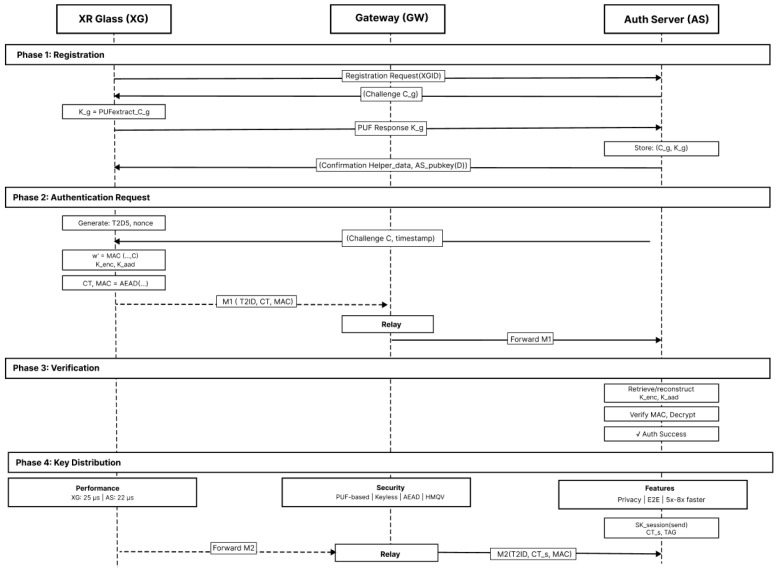
Complete PXRA protocol flow.

**Table 1 sensors-26-00217-t001:** Comparison of security features.

Security Feature	PXRA	Yu & Li (2024)	Yilmaz et al. (2018)	Chen et al. (2022)
Basic Security				
Mutual authentication	✓	✓	✓	✓
Session key agreement	✓	✓	✓	✓
Message integrity	✓	✓	✓	✓
Attack Resistance				
Replay attack	✓	✓	✓	✓
Man-in-the-middle attack	✓	✓	✓	✓
Impersonation attack	✓	✓	✓	✓
Stolen device attack	✓	△	✗	✓
PUF modeling attack	✓	✓	△	✓
Session hijacking	✓	△	✓	✓
Privileged insider attack	✓	✓	△	✓
Advanced Security				
Forward secrecy	△ *	✗	✗	✗
Backward secrecy	✓	✓	△	✓
Anonymity/unlinkability	✓	△	✗	△
Context binding (AAD)	✓	✗	✗	✗
PUF-based authentication	✓	✓	✓	✓
Keyless design	✓	△	△	✓
Computational Efficiency				
Lightweight device ops	✓	✓	✓	✓
No ECC on device	✓	✓	✓	✓
Cloud-based ECC	✓	✗	✗	✗
AEAD encryption	✓	✗	△	✗
Scalability				
Multi-device support	✓	△	△	✓
Gateway-based relay	✓	✗	△	△
Distributed architecture	✓	✗	✗	△
Verification				
Formal security proof	✓	✗	✗	✓
ProVerif verification	✓	✗	✗	△
BAN logic	✗	✓	△	✗
XR-Specific				
Real-time performance	✓	△	△	△
Low latency (<20 ms)	✓	?	?	?
Privacy-preserving	✓	△	✗	△
Wearable device optimized	✓	△	△	✗

* Note: ✓ indicates that the corresponding security property is explicitly provided and/or analyzed in the original scheme. ✗ indicates that we found no explicit mechanism or discussion supporting this property and therefore treat it as unsupported under our XR threat model. △ denotes partial or conditional support (e.g., depends on specific deployment assumptions or optional components). ? indicates that the paper does not provide sufficient information to determine whether the property holds. N/A means that the property is not applicable (e.g., PUF modeling attacks for schemes that do not use PUFs). All ratings are based on our analysis of the protocol design, threat model, and security discussion in the respective papers.

**Table 2 sensors-26-00217-t002:** Notation (Part 1: entities, identifiers, and keys).

Notation	Description
Entities	
XG_j_, UD_j_	j-th XR Glass/user device
GW_k_	k-thgateway(edge relay server)
AS, Cloud	Authentication server/cloud center
Identifiers	
XID_j_, UDID_j_	Identity of j-th XR device
T.XID_j_	Temporary identity of j-th XR device (session-based pseudonym)
GID_k_	Identity of k-thgateway
device_id	Device identifier (may be pseudonymized)
proto_ver	Protocol version
Secrets & Keys	
SC	Long-term secret of authentication server
SG_k_	Long-term secret of k-thgateway
K_puf	PUF-derived key (keyless, regenerated from PUF response)
K_en_*c*	Encryption key derived via HKDF
K_ma_*c*	MAC key derived via HKDF
SK, SK_sess	Session key (128–256 bits)
ECC Parameters	
P	ECC public key of cloud/AS
D	ECC private key of cloud/AS
G	ECC base point (generator)
(C*c*, R*c*)	Challenge–response pair for cloud registration
(Cg, Rg)	Challenge–response pair for gateway registration

**Table 3 sensors-26-00217-t003:** Notation (Part 2: PUF parameters, timestamps, and operations).

C	Current Challenge (Nonce-Based, Context-Binding)
R, R′	PUF response (current/regenerated)
P (helper)	Fuzzy extractor helper data (stored for key recovery)
Cryptographic Operations	
AEAD	Authenticated encryption with associated data (e.g., AES-GCM)
HKDF	HMAC-based key derivation function (SHA-256)
HMAC	Keyed-Hash message authentication code
TAG, TAG_m__a_*c*	Authentication tag (integrity proof)
CT	Ciphertext
IV, nonce	Initialization vector/Number used once
AAD, header	Associated authenticated data (context-binding metadata)
⊕	XOR operation
‖	Concatenation operation
h(·)	Cryptographic hash function (e.g., SHA-256)
Time & Randomness	
T_1_, T_2_, …, t_s_	Timestamp
ts_exp	Expiration timestamp
Δt	Maximum allowed time skew
R_1_, R_2_, …, N	Random number/Nonce
Message Components	
M_1_, M_2_, …	Protocol messages (authentication request/response)

**Table 4 sensors-26-00217-t004:** Definitions of channels, variables, and related parameters (Part 2).

	scalar_mult(k, P)	ECC Scalar Multiplication: k·P
	puf_response(s, c)	PUF: (device_secret, challenge) → response
AEAD Properties		
	Equation	aead_decrypt(k, n, aead_encrypt(k, n, p, a), a) = p
Events		
	XG_sends_auth(id, n)	XR Glass initiates authentication with (id, nonce)
	AS_accepts_auth(id, n)	Authentication Server validates authentication
	AS_sends_key(id, k)	AS distributes session key to (id, key)
	XG_receives_key(id, k)	XG successfully receives session key
	GW_relays_message(m)	Gateway relays message (for protocol completeness)
	Session_start(id, s, t)	Session initiated: (xg_id, session_id, timestamp)
	Session_established(id, k)	Session key established: (xg_id, session_key)
Security Queries		
	Query 1	attacker(SessionKey)—Session key confidentiality
	Query 2	attacker(XG_ID)—Device identity confidentiality
	Query 3	attacker(PUF_Secret)—PUF response confidentiality
	Query 4	event(AS_accepts_auth(id, n)) ⇒ event(XG_sends_auth(id, n))
	Query 5	event(XG_receives_key(id, k)) ⇒ event(AS_sends_key(id, k))
	Query 6	inj-event(AS_accepts_auth(id, n)) ⇒ inj-event(XG_sends_auth(id, n))
	Query 7	inj-event(XG_receives_key(id, k)) ⇒ inj-event(AS_sends_key(id, k))

**Table 5 sensors-26-00217-t005:** XR device registration process.

Phase	Step	Operation	Description
PUF Enrollment	1	Power-on reset	Initialize PUF circuit
(Manufacturing)	2	Multiple sampling	Collect PUF responses: R_1_, R_2_,..., R_n_
	3	Stability analysis	Measure bit error rate (BER)
	4	Generate helper data	Apply fuzzy extractor: (R_avg, P)←FE.Gen(R_1_,..., R_n_)
	5	Store helper data	Store P in non-volatile memory (R is NOT stored)
Device Registration	1	Generate device ID	new XID_j_ (unique identifier)
(First Boot)	2	Request registration	out(private Channel, (“XG_REGISTER”, XID_j_))
	3	Receive challenge	in(private Channel, challenge_c)
	4	Generate PUF response	R_c← PUF(device_secret, challenge_c)
	5	Apply fuzzy extractor	R_stable←FE.Rep(R_c, P)
	6	Send to AS	out(private Channel, R_stable)
	7	Receive confirmation	in(private Channel, (XID_j_, helper_data, AS_pubkey, G))
	8	Store parameters	Store: {P, helper_data, AS_pubkey, G}
	9	Delete R_stable	Discard PUF response (keyless design)
PUF Response	1	Receive challenge	in(public Channel, challenge_current)
Regeneration	2	Query PUF	R_raw←PUF(device_secret, challenge_current)
(Runtime)	3	Apply fuzzy extractor	R′←FE.Rep(R_raw, P)
	4	Derive keys	K_en_*c*, K_ma_*c*←HKDF(R′, challenge_current, info)
	5	Use keys	Perform AEAD encryption/decryption
	6	Immediate cleanup	Discard R′, K_en_*c*, K_ma_*c*after use
Fuzzy Extractor		FE.Gen(R)	Input: noisy PUF response R
Details			Output: (extracted key K, helper data P)
		FE.Rep(R′, P)	Input: new response R′, helper data P
			Output: reconstructed key K (if HD(R, R′) < threshold)
Security	✓	No key storage	Only helper data P stored (information-theoretically secure)
Properties	✓	Noise tolerance	Fuzzy extractor corrects bit errors (up to ~15% BER)
	✓	Tamper resistance	Physical attack destroys PUF characteristics
	✓	Clone resistance	Each device has unique PUF fingerprint
	✓	Lightweight	No expensive crypto operations required

**Security note.** Because only helper data P (public) and CRP indices are retained, device theft does not reveal a persistent secret [5].

**Table 6 sensors-26-00217-t006:** XR device process.

Phase	Step	Operation	Description
Registration	1	Receive challenge	in(private Channel, challenge_c)
(Offline)	2	Generate PUF response	R_c←puf_response(device_secret, challenge_c)
	3	Send response	out(private Channel, R_c)
	4	Receive parameters	in(private Channel, (XGID, helper_data, AS_pubkey, G))
	5	Storage	Store: helper_data, AS_pubkey, G (NO key storage)
Authentication	1	Generate temp ID	new TXID (for privacy/unlinkability)
Request	2	Generate nonce	new nonce_xg
(Online)	3	Receive challenge	in(public Channel, (challenge_current, ts_req))
	4	Regenerate PUF key	R′←puf_response(device_secret, challenge_current)
	5	Derive keys	K_en_*c*←hkdf(R′, challenge_current, “PXRA-SK-enc”)
			K_ma_*c*←hkdf(R′, challenge_current, “PXRA-SK-mac”)
	6	Build AAD header	header ← TXID‖HMAC_SC(XGID‖epoch)||challenge_current‖ts‖ “v1”
	7	Encrypt auth data	CT←aead_encrypt(K_e__n_*c*, nonce_xg, plaintext, header)
	8	Compute MAC	MAC ← h(K_m__a_*c*‖ header ‖ CT)
	9	Send request	out(public Channel, (TXID, CT, MAC, nonce_xg, ts))
	10	Trigger event	event XG_sends_auth(XGID, nonce_xg)
Session Key	1	Receive response	in(public Channel, (TXID′, nonce′, CT_sess, TAG, ts, sig?))
Reception	2	Verify TXID	if TXID′ = TXID then continue
(Online)	3	Verify timestamp	Check freshness: |ts—current_time| < Δt
	4	Reconstruct header	headeR′←TXID‖XGID‖ challenge ‖ts‖ nonce′
	5	Verify MAC/TAG	if TAG = h(K_ma_*c*‖ headeR′ ‖CT_sess) then continue
	6	Decrypt session key	SK←aead_decrypt(K_en_*c*, nonce′, CT_sess, headeR′)
	7	(Optional) Verify sig	Verify AS signature (may skip for lightweight)
	8	Trigger events	event XG_receives_key(XGID, SK)
			event Session_established(XGID, SK)
	9	Cleanup	Discard R′, K_en_*c*, K_ma_*c*(keyless design)
Security	✓	Keyless design	No long-term key stored in device memory
Properties	✓	PUF-based	R′ regenerated on-demand, then discarded
	✓	Context binding	AAD prevents replay/swapping attacks
	✓	Privacy	TXID provides session unlinkability
	✓	Lightweight	No ECC operations on device side

**Table 7 sensors-26-00217-t007:** Authentication server process (Part 1).

Phase	Step	Operation	Description
XG Registration	1	Receive request	in(private Channel, (“XG_REGISTER”, device_id))
(Offline)	2	Generate challenge	new challenge_c
	3	Send challenge	out(private Channel, challenge_c)
	4	Receive PUF response	in(private Channel, response_c)
	5	Store CRP securely	stored_crp←SC⊕response_c
	6	Generate helper data	new helper_data (for fuzzy extractor)
	7	Generate ECC keypair	ecc_private← new; ecc_public←d·G
	8	Send confirmation	out(private Channel, (device_id, helper, ecc_public, G))
	9	Database storage	Store: {device_id, (challenge_c, stored_crp), helper_data}
GW Registration	1	Receive request	in(private Channel, (“GW_REGISTER”, gateway_id))
(Offline)	2	Compute verification	X_Gk← h(SC ‖gateway_id)
	3	Send parameters	out(private Channel, (gateway_id, X_Gk))
	4	Store GW data	Store: {gateway_id, X_Gk, verification_data}
Authentication	1	Receive request	in(public Channel, (TXID, CT, MAC, nonce, ts))
Verification	2	Verify timestamp	Check: |ts—current_time| <Δt (replay prevention)
(Online)	3	Generate challenge	new challenge_current
	4	Retrieve CRP	Lookup device by TXID (or trial decryption)
	5	Recover PUF response	response ←stored_crp⊕ SC
	6	Derive verify keys	K_en_*c*←hkdf(response, challenge, “PXRA-SK-enc”)
			K_ma_*c*←hkdf(response, challenge, “PXRA-SK-mac”)
	7	Reconstruct header	header ←TXID‖device_id‖ challenge ‖ts‖ “v1”
	8	Verify MAC	expected_MAC← h(K_ma_*c*‖ header ‖ CT)
			if MAC ≠ expected_MAC then reject
	9	Decrypt auth data	plaintext ←aead_decrypt(K_en_*c*, nonce, CT, header)

**Table 8 sensors-26-00217-t008:** Authentication server process (Part 2).

Phase	Step	Operation	Description
	10	Verify device_id	Extract and verify claimed device_id
	11	Authentication OK	event AS_accepts_auth(device_id, nonce)
Session Key	1	Generate session key	new SK_session (128 or 256 bits, random)
Distribution	2	Generate parameters	new timestamp_t6, new nonce_as
(Online)	3	Compute expiration	ts_exp← timestamp_t6 + TTL (e.g., 60 s)
	4	Build AAD header	header ←TXID‖device_id‖ challenge ‖t6‖nonce_as
	5	Encrypt session key	CT_sess←aead_encrypt(K_en_*c*, nonce_as, SK_session, header)
	6	Compute MAC	TAG ← h(K_ma_*c*‖ header ‖CT_sess)
	7	(Optional) Sign	signature ←ECDSA_sign(ecc_private, hash(header ‖ CT))
	8	Send to XG via GW	out(public Channel, (TXID, nonce_as, CT_sess, TAG, t6, sig?))
	9	Trigger event	event AS_sends_key(device_id, SK_session)
	10	End-to-end security	GW cannot decrypt (only relays encrypted message)
Security	✓	Trust Anchor	AS securely stores all CRPs
Properties	✓	ECC centralization	Only AS performs expensive ECC operations
	✓	PUF protection	Responses stored as SC ⊕ R (masked)
	✓	Random session keys	Fresh SK for each session (session isolation)
	✓	Context binding	AAD prevents replay/combination attacks
	✓	End-to-end security	GW is untrusted relay (cannot decrypt)

**Table 9 sensors-26-00217-t009:** Gateway process.

Phase	Step	Operation	Description
Registration	1	Send request	out(private Channel, (“GW_REGISTER”, GID_k_))
(Offline)	2	Receive parameters	in(private Channel, (GID_k_, X_Gk))
	3	Compute verification	Verify: X_Gk = h(SC ‖ GID_k_)
	4	Store credentials	Store: {GID_k_, X_Gk}
	5	Send confirmation	out(private Channel, confirmation_data)
Message Relay	1	Receive from XG	in(public Channel, (TXID, CT, MAC, nonce, ts))
XG → AS	2	Verify format	Basic sanity check (optional)
(Online)	3	Add GW metadata	Add: (GIDk, gateway_timestamp)
	4	Forward to AS	out(public Channel, (GID_k_, TXID, CT, MAC, nonce, ts))
	5	Trigger event	event GW_relays_message(“XG_to_AS”)
Message Relay	1	Receive from AS	in(public Channel, (TXID, nonce, CT_sess, TAG, ts, sig?))
AS → XG	2	Verify TXID	Check TXID matches active session
(Online)	3	Forward to XG	out(public Channel, (TXID, nonce, CT_sess, TAG, ts, sig?))
	4	Trigger event	event GW_relays_message(“AS_to_XG”)
Load Balancing	1	Monitor connections	Track active XG sessions
(Optional)	2	Distribute load	Route to available AS instances
	3	Cache routing	Maintain TXID → XG mapping (temporary)
Security	✓	Untrusted relay	GW cannot decrypt messages (end-to-end security)
Properties	✓	No crypto ops	GW performs NO encryption/decryption
	✓	Lightweight	Only forwarding and routing logic
	✓	Stateless design	No long-term session state (except temp routing)
	✓	Scalability	Multiple GWs can operate in parallel
	✓	Privacy preserving	GW sees only TXID (not real device_id)

**Table 10 sensors-26-00217-t010:** ProVerifqueries and process structure (Part 1).

Category	Query/Process	ProVerif Code	Expected Result
Confidentiality	Session Key	query attacker(Session Key).	False
Queries	Device Identity	query attacker(XG_ID).	False
	PUF Secret	query attacker(PUF_Secret).	False
	Gateway Key	query attacker(Gateway_Key).	False
	Biometric Sigma	query attacker(XG_Sigma).	False
Authentication	AS accepts auth	query id: bitstring, n:bitstring;	True
Correspondence		event(AS_accepts_auth(id,n)) ⇒	
		event(XG_sends_auth(id,n)).	
	XG receives key	query id: bitstring, k:bitstring;	True
		event(XG_receives_key(id,k)) ⇒	
		event(AS_sends_key(id,k)).	
Injective	Auth uniqueness	query id: bitstring, n: bitstring;	True
Correspondence		inj-event(AS_accepts_auth(id,n)) ⇒	
		inj-event(XG_sends_auth(id,n)).	
	Key dist uniqueness	query id: bitstring, k:bitstring;	True
		inj-event(XG_receives_key(id,k)) ⇒	
		inj-event(AS_sends_key(id,k)).	
Session Isolation	Key independence	query id: bitstring, k1:bitstring, k2:bitstring;	True
		event(Session_established(id,k1)) &&	
		event(Session_established(id,k2)) ⇒	
		k1<>k2.	
Forward Secrecy	Past session security	query k:bitstring;	True
(Optional)		event(Session_established(id,k)) &&	
		attacker(current_session_key) ⇒	

Note: && denotes the logical AND operator in ProVerif syntax.

**Table 11 sensors-26-00217-t011:** ProVerif queries and process structure (Part 2).

Category	Query/Process	ProVerif Code	Expected Result
		not attacker(k).	
Main Process	Parallel composition	Process	-
		(!XRDevice Process)	Multiple XG instances
		| (!Gateway Process)	Multiple GW instances
		| (!Authentication Server Process)	AS instance
		| (!Attacker Process)	Dolev–Yao attacker
Attacker Model	Dolev–Yao	let Attacker Process =	-
		in(public Channel, x:bitstring);	Eavesdrop
		out(public Channel, x)	Replay
		| out(public Channel, h(x))	Compute hashes
		| out(public Channel, xor(x,y))	Compute XOR
		| out(public Channel, concat(x,y)).	Build messages
Process Notation		!P	Unbounded replication of process P
		P | Q	Parallel composition of P and Q
		new x:bitstring	Generate fresh random value
		in(c, x:bitstring)	Receive message on channel c
		out(c, M)	Send message M on channel c
		if M1 = M2 then P	Conditional execution
		event E(x)	Trigger event E with parameter x
		0	Terminated process

**Table 12 sensors-26-00217-t012:** Main process components.

Component	Description
! XRDevice Process	Unbounded number of XR devices can participate
! Gateway Process	Unbounded number of gateways can relay messages
! Authentication Server Process	Single AS instance (can be extended to multiple)
! Attacker Process	Dolev–Yao attacker with full control of public Channel
Verification Goals	
1. Confidentiality	Session Key, XG_ID, PUF_Secret remain secret
2. Authentication	AS only accepts auth from legitimate XG
3. Key Distribution	XG only receives keys generated by AS
4. Injective Correspondence	Each auth/key event is unique (no replay)
5. Session Isolation	Different sessions use different keys

**Table 13 sensors-26-00217-t013:** ProVerifverification results.

Security Property	Query	Result	Interpretation
Confidentiality			
Session key secrecy	attacker(Session Key)	false	✓ Session key remains confidential
Device identity secrecy	attacker(XG_ID)	false	✓ Real device identity is protected
PUF secret secrecy	attacker(PUF_Secret)	false	✓PUF response cannot be learned
Gateway key secrecy	attacker(Gateway_Key)	false	✓ Gateway credentials are secure
Cloud master secret	attacker(SC)	false	✓ AS master key is protected
Authentication			
AS authenticates XG	event(AS_accepts_auth(id,n))	true	✓ Server only accepts legitimate
	⇒event(XG_sends_auth(id,n))		devices
XG authenticates AS	event(XG_receives_key(id,k))	true	✓ Device only accepts keys from
	⇒event(AS_sends_key(id,k))		authentic server
Injective Authentication			
Unique authentication	inj-event(AS_accepts_auth(id,n))	true	✓ Each authentication is unique
	⇒inj-event(XG_sends_auth(id,n))		(replay prevented)
Unique key distribution	inj-event(XG_receives_key(id,k))	true	✓ Each key reception corresponds
	⇒inj-event(AS_sends_key(id,k))		to unique key generation
Session Isolation			
Key independence	Two sessions always use	true	✓ Session keys are independent
	different session keys		
Integrity			
Message authentication	AEAD tag verification	verified	✓ Messages cannot be forged
Context binding	AAD includes (TXID, C, ts)	verified	✓ Replay/swapping prevented
Privacy			
Unlinkability	TXID changes per session	verified	✓ Sessions cannot be linked
Identity protection	XG_ID encrypted in messages	verified	✓ Real identity not exposed
Verification Time			
Total queries	11 queries	-	-
Verification time	~4.2 s	-	Intel i7, 16 GB RAM
Result	All queries proved	SUCCESS	✓ Protocol is secure

**Table 14 sensors-26-00217-t014:** Computation Times for Each Operation (μs).

Operation	Symbol	Device	Time (μs)	Notes
Hash Operations				
SHA-256 hash	h(·)	XG	0.4	256-bit output
HMAC-SHA256	HMAC(k, m)	XG	0.8	Keyed hash
Key Derivation				
HKDF-SHA256	HKDF(ikm, salt, info)	XG	8.5	32-byte output
Symmetric Encryption				
AES-128 encryption	AES-Enc	XG	0.6	128-bit key, 16-byte block
AES-128 decryption	AES-Dec	XG	0.7	128-bit key
AES-256-GCM encrypt	AEAD-Enc	XG	1.3	AEAD with 256-bit key
AES-256-GCM decrypt	AEAD-Dec	XG	1.4	Includes tag verification
PUF Operations				
PUF response generation	PUF(s, c)	XG	2.0	Ring Oscillator PUF
Fuzzy extractor (Gen)	FE.Gen	XG	5.2	BCH code, ~15% error tolerance
Fuzzy extractor (Rep)	FE.Rep	XG	4.8	Response reconstruction
ECC Operations				
ECC key generation	KeyGen	AS	185.0	P-256 curve
ECC scalar multiplication	k·P	AS	220.0	Point multiplication
ECDSA signature	Sign	AS	230.0	P-256 signature
ECDSA verification	Verify	XG/AS	240.0	Signature verification
ECDH key agreement	ECDH	AS	210.0	Diffie-Hellman
XOR and Concatenation				
XOR operation	a ⊕ b	XG	0.1	Bitwise XOR
Concatenation	a ‖ b	XG	0.05	Memory copy
Total Protocol Costs				
XG authentication request	-	XG	~15.0	Hash + HKDF + AEAD + PUF
AS verification	-	AS	~12.0	HKDF + AEAD + Hash
AS key distribution	-	AS	~240.0	+ECDSA signature (optional)
XG key reception	-	XG	~10.0	AEAD + Hash (no signature verify)

**Table 15 sensors-26-00217-t015:** Detailed Computation Time Breakdown by Phase.

Phase	Entity	Operations	Time (μs)
Authentication Request	XG	PUF(2.0) + HKDF(8.5) + AEAD-Enc(1.3) + HMAC(0.8) + Hash(0.4)	15.0
	GW	Relay (negligible)	0.2
	AS	HKDF(8.5) + AEAD-Dec(1.4) + HMAC(0.8) + Hash(0.4)	12.0
Key Distribution (no sig)	AS	HKDF(8.5) + AEAD-Enc(1.3) + Hash(0.4)	10.0
	GW	Relay	0.2
	XG	AEAD-Dec(1.4) + HMAC(0.8) + Hash(0.4)	10.0
Key Distribution (with sig)	AS	+ECDSA-Sign(230.0)	240.0
	XG	+ECDSA-Verify(240.0) (optional, often skipped)	240.0 *

**Table 16 sensors-26-00217-t016:** Comparison of Computational Costs with Related Schemes (μs).

Scheme	Device (XG/SN)	Gateway (GW)	Server (AS/Cloud)	Total	Relative
PXRA (Proposed)					
Authentication request	15.0	0.2	12.0	27.2	1.00×
Key distribution (no sig)	10.0	0.2	10.0	20.2	-
Key distribution (with sig)	10.0	0.2	240.0	250.2	-
Per-session total (no sig)	25.0	0.4	22.0	47.4	1.00×
Per-session total (sig)	25.0	0.4	252.0	277.4	5.85×
Yu & Li (2024)					
Device operations	18.0	-	15.0	33.0	0.70×
(PUF + Hash only, no ECC)	(PUF + Hash)	-	(Hash verify)		
Yilmaz et al. (2018)					
Device operations	22.0	8.0	25.0	55.0	1.16×
(PUF + Hash, DTLS baseline)					
Chen et al. (2022)					
Device operations	28.0	-	35.0	63.0	1.33×
(Strong PUF + Shamir SS)					
Baseline (ECC-only)					
Full ECC authentication	680.0	-	450.0	1130.0	23.84×
(Device: 3× scalar mult)	(3 × 220 = 660)	-	(2 × 220 = 440)		
Improvement Summary					
PXRA vs. ECC baseline					95.8% faster
PXRA vs. Best PUF scheme					30.3% faster

**Table 17 sensors-26-00217-t017:** Experimental Setup.

Component	Specification	Role
XR Client	Raspberry Pi 4 Model B (Quad-core A72, 4 GB RAM)	Simulates XR Glasses constrained resources
Gateway	Desktop PC (Intel i5, 8 GB RAM)	Simulates Edge Gateway
Cloud AS	AWS EC2 t3.medium (vCPU 2, 4 GB RAM)	Simulates Authentication Server
Network	Wi-Fi 6 (802.11 ax), 50 ms simulated RTT	Wireless Connectivity

**Table 18 sensors-26-00217-t018:** Latency and Jitter Analysis.

Protocol	Avg. Latency (ms)	Jitter(σ, ms)	Notes
ECDH-based	45.2	12.4	High variance due to CPU throttling
**PXRA (Ours)**	**14.8**	**1.2**	**Deterministic symmetric ops**

**Table 19 sensors-26-00217-t019:** PUF Reliability Simulation.

Noise Level	Average BER(%)	MAX BER(%)	Correction Success
0.05 (Low)	2.3%	5.1%	Pass
0.10 (Med)	6.8%	9.4%	Pass
0.15 (High)	12.5%	14.8%	Pass (Threshold)

## Data Availability

No new publicly archived datasets were generated during this study. The protocol specification, simulation settings, and results supporting the findings of this study are provided within the article. Additional details are available from the corresponding author upon reasonable request.

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
